# Melatonin as a Coadjuvant in the Treatment of Patients with Fibromyalgia

**DOI:** 10.3390/biomedicines11071964

**Published:** 2023-07-12

**Authors:** David González-Flores, Laura López-Pingarrón, María Yolanda Castaño, María Ángeles Gómez, Ana B. Rodríguez, Joaquín J. García, María Garrido

**Affiliations:** 1Department of Anatomy, Cell Biology and Zoology, Science Faculty, University of Extremadura, 06006 Badajoz, Spain; 2Neuroimmunophysiology and Chrononutrition Research Group, University of Extremadura, 06006 Badajoz, Spain; ycblanco@unex.es (M.Y.C.); magomez@unex.es (M.Á.G.); moratino@unex.es (A.B.R.); mgaalvarez@unex.es (M.G.); 3Oxidative Stress and Aging Research Group, Department of Pharmacology, Physiology, Legal and Forensic Medicine, University of Zaragoza, 50009 Zaragoza, Spain; jjgarcia@unizar.es; 4Department of Nursing, Merida University Center, University of Extremadura, 06006 Badajoz, Spain; 5Department of Physiology, Science Faculty, University of Extremadura, 06006 Badajoz, Spain

**Keywords:** antioxidant, antioxidant capacity, fibromyalgia, melatonin, sleep

## Abstract

Fibromyalgia syndrome (FMS) is a chronic widespread pain syndrome that is accompanied by fatigue, sleep disturbances, anxiety, depression, lack of concentration, and neurocognitive impairment. As the currently available drugs are not completely successful against these symptoms and frequently have several side effects, many scientists have taken on the task of looking for nonpharmacological remedies. Many of the FMS-related symptoms have been suggested to be associated with an altered pattern of endogenous melatonin. Melatonin is involved in the regulation of several physiological processes, including circadian rhythms, pain, mood, and oxidative as well as immunomodulatory balance. Preliminary clinical studies have propounded that the administration of different doses of melatonin to patients with FMS can reduce pain levels and ameliorate mood and sleep disturbances. Moreover, the total antioxidant capacity, 6-sulfatoxymelatonin and urinary cortisol levels, and other biological parameters improve after the ingestion of melatonin. Recent investigations have proposed a pathophysiological relationship between mitochondrial dysfunction, oxidative stress, and FMS by looking at certain proteins involved in mitochondrial homeostasis according to the etiopathogenesis of this syndrome. These improvements exert positive effects on the quality of life of FMS patients, suggesting that the use of melatonin as a coadjuvant may be a successful strategy for the management of this syndrome.

## 1. Introduction

The history of fibromyalgia (FMS) as a pathological entity is fairly recent. During the 19th century, it was known as fibrositis, nodular fibromyositis, or psychogenic rheumatism, and it was not until the 1970s that it received an exhaustive clinical description. In 1992 it was recognized by the World Health Organization (WHO), and in 1994 by the International Association for the Study of Pain, with the code X33.X8a. FMS is now included as a diagnosis in the tenth revision of the WHO International Statistical Classification of Diseases and Health-Related Problems, as part of ‘Other soft tissue disorders, not specified elsewhere’ [1,2], though it could be considered as a range of disorders with a common clinical expression and variable symptom intensity [3].

The prevalence of FMS is estimated at 2–8% in the world population and 2.4% in Spain [2]. FMS can develop at any age, though it usually appears between 20 and 50 years old, but it has also been detected in childhood. FMS has a higher incidence in women than in men, with a ratio of 10:1 [2]. Low education and socioeconomic levels are considered risk factors for the development of the disease. The presence of comorbidities, such as headaches or migraines, chronic fatigue syndrome, and irritable bowel syndrome, is very common in FMS [1].

Only 12–28% of patients identified during population-based surveys who met the 1990 criteria of the American College of Rheumatology (ACR) were diagnosed with FMS [1]. The diagnosis of fibromyalgia was proposed in the 90s, taking into account the findings of tender points that a doctor notes when exploring the symptoms that a patient refers to (pain upon pressure at certain points of the body without other alterations); however, in 2010 two questionnaires provided by patients, a generalized pain index (the Widespread Pain Index (WPI)) and a severity scale of symptoms (the Symptom Severity Score (SS-Score)), were proposed as new diagnostic criteria [3]. Taking everything into account, the majority of studies have shifted their focus from etiology to symptom management.

Recently, several studies have shown that oxidative stress is involved in the pathophysiology of FMS [4,5]. In patients with FMS, plasma levels of lipid peroxidation are increased, reflecting the intracellular production of reactive oxygen species (ROS) [6]; levels of carbonylated proteins are also high as the end products of membrane damage induced by ROS [7,8]. However, a decrease in the total antioxidant capacity or antioxidant enzymes, such as catalase or superoxide dismutase, has also been observed [7,8,9], as has an association between plasma levels of toxic heavy metals and essential metals with oxidative stress markers [10]. An alteration in the plasma distribution of coenzyme Q10 (CoQ10), which leads to respiratory chain dysfunction by affecting mitochondrial uncoupling proteins, and higher levels of ROS production in mononuclear cells have also been demonstrated in FMS patients [11].

ROS are involved in the etiology of pain, an important symptom in FMS [12]. The pathophysiological link between mitochondrial dysfunction and oxidative stress is still unknown, but the etiogenesis of FMS would be consistent with it [13]. The loss of mitofusin 2 (Mfn2), an outer mitochondrial membrane protein that mediates mitochondrial fusion, has been suggested by several studies [14,15,16] to cause the depletion of CoQ10, altered mitochondrial permeability transition pores, and ROS production [17]. Furthermore, CoQ10 controls serotonin levels and depressive symptoms in FMS patients [18]. The role Mfn2 plays in regulating CoQ10 for optimal mitochondrial respiratory chain performance is unclear, but a surprising and novel role of Mfn2 in maintaining the terpenoid biosynthesis pathway has been shown, which is required for mitochondrial CoQ10 production [17]. In addition, other studies have shown that the peroxisome proliferator-activated receptor gamma coactivator-1alpha (PGC-1) signaling pathway can regulate Mfn2 gene and protein expression [19].

FMS is defined as a chronic disease of unknown etiology and is part of a variety of syndromes that have neither precise classifications nor specific tests that allow their diagnosis. It has a large impact on the quality of life of patients, and, in some cases, is a disabling situation [2,20]. Although this syndrome has been considered a psychosomatic disorder, recent studies propose that FMS is a disorder of pain control at the brain level, so it could be classified within the central sensitization syndromes [2]. As FMS is an extremely complex syndrome with a variable symptom landscape and still unknown etiology, its pharmacological treatment is generally focused on the relief of anxiety and pain, increasing the mood/emotional state of the patient, or combating nighttime sleep disturbances and/or insomnia. In several mood disorders, such as seasonal affective, bipolar, and major depression, disturbances of sleep and circadian rhythms are core symptoms [21]. Melatonin, a neurohormone mainly synthesized by the pineal gland, in a circadian pattern, reaching peak levels during the night and under the control of suprachiasmatic nucleus [22], is both a marker and regulator of circadian rhythms and sleep [21]. Alterations in melatonin synthesis and the expression of the MT1 and MT2 receptors have been reported in patients with mood disorders, e.g., the MT1 receptor is associated with anxiety-like and obsessive–compulsive-like behaviors, and the MT2 melatonin receptor is associated with depressive-like behavior and may play a role in the pathology of major depression [21,23]. The drugs that are used in FMS are antidepressants, anti-inflammatories, muscle relaxants, and sedative hypnotics. Studies combining several drugs (milnacipram + pregabalin or paracetamol + tramadol) improved the symptoms of the disease [24,25]. Melatonin alone or associated with antidepressants may be involved in modifying the endogenous pain-modulating system in fibromyalgia [26]. One such strategy involves targeting melatonin receptors, as melatonin has a key role in synchronizing circadian rhythms, which are known to be perturbed in depressed states, and therefore may also be considered in the therapeutic approach to FMS [27,28].

## 2. Role of Melatonin in Fibromyalgia

Melatonin is a low-molecular-weight indolamine (N-acetyl-5-methoxytryptamine) synthesized from the amino acid L-tryptophan, which is present in animals, plants, and even unicellular organisms. Melatonin is produced in the main cell of the pineal gland, the pinealocyte, and is controlled in mammals via ambient light. Despite having a simple structure, melatonin has been shown to be a powerful endogenous antioxidant [29]. In addition, it is involved in the control of reproduction [30], immunomodulation [31], and biological rhythms [32], improves mood [33], and possesses anti-inflammatory effects [34,35].

### 2.1. Receptor-Mediated Effects of Melatonin and Its Involvement in Fibromyalgia

Many of the actions of melatonin are mediated by G-protein-coupled receptors, MT1 and MT2. It is also known that melatonin binds to the quinone reductase II enzyme, previously defined as the MT3 receptor. Functionally, MT1 and MT2 have distinctive physiological roles. For example, MT1 mediates melatonin-regulated cardiac vasoconstriction [36,37], whereas MT2 activation dilates cardiac vessels and modulates inflammatory as well as immune responses [38]. A more recent study showed that MT1 may also modulate biological-clock-related gene expression, as the expression of most clock genes is reduced in the pituitary of MT1 knockout mice but not in MT2 knockout mice [39]. Melatonin apparently acts as a natural ligand for the retinoid-related orphan nuclear hormone receptor family [40,41]. The immunomodulatory effects and possibly also a portion of the circadian effects are partially mediated through the melatonin activation of nuclear receptors that repress 5-lipoxygenase mRNA expression in human B cell lines [42]. In addition, melatonin may be activated by binding to intracellular proteins such as calmodulin, which participates in second messenger signal transduction; this directly antagonizes calmodulin binding to Ca^2+^ [43]. Finally, melatonin has important antioxidative effects, which have been extensively investigated in various pathological conditions associated with free radicals and related reactants, such as ischemia/reperfusion, inflammation, ionizing radiation, and mitochondrial toxins [44,45]. It has been suggested that melatonin also potentiates free radical scavenging through a nonenzymatic process of electron donation [46,47].

### 2.2. Interaction of Melatonin with Receptors Implicated in the Pain in FMS

As both melatonin secretion and pain perception follow a circadian rhythm, many researchers have confirmed that melatonin plays a substantial role in the regulation of pain under physiological conditions [48,49]. For example, melatonin MT2 receptor agonists require mu opioid receptor (MOR) activation to exert their antiallodynic effects, i.e., a pain response to a non-noxious stimulus, involving MORs and MT2 receptors through the modulation of descending antinociceptive pathways in the periaqueductal gray of the brainstem [50,51].

β-endorphins, gamma-aminobutyric acid (GABA) receptors, the nitric oxide (NO)–arginine pathway, and opioid 1 receptors may all be involved in melatonin-mediated analgesic actions. Melatonin may modulate the function of GABA receptors [52], increasing GABA concentration by 50% [53]. Melatonin increases the release of β-endorphins from the pituitary gland, the antagonist of which may also inhibit melatonin-induced antinociceptive effects [54,55]. Furthermore, melatonin-induced long-term analgesia may be antagonized by naloxone [56]. In fact, the expression of MOR mRNA follows a circadian pattern, where MORs are more expressed during the late light phase and less during the dark phase. In the MT2^−/−^ knockout mice, the lack of an MT2 endogenous tone might activate neuronal compensatory mechanisms through increased *Penk* mRNA expression in the rostral ventromedial medulla (RVM), leading to the upregulation of the endogenous opioid encephalin at the central level involved in the modulation of pain. These findings may corroborate the hypothesis that the MT2 receptor plays a specific role in nociception, particularly during the inactive phase (day), when MT2 is more abundant in the brain. Thus, the increased sensitivity during the night of MT2^−/−^ might be related to the scarce availability of MORs in these areas of the descending antinociceptive pathway [57]. Melatonin has been found to reduce inflammatory pain, probably by blocking the production of NO by inducible NO synthase and the signaling of NO-cyclic GMP [58,59]. Melatonin may also mediate its analgesic activity by interacting with benzodiazepinergic, muscarinic, nicotinic, serotonergic, and α_1_ as well as α_2_ adrenergic receptors located in the central nervous system and the dorsal horn of the spinal cord. Its antinociceptive effects can also be produced by affecting the sigma system, dopaminergic receptor, and glutamatergic receptor (NMDA type) [60,61]. Melatonin’s efficacy as an analgesic and anxiolytic drug has been demonstrated in various animal models of pain, leading to its therapeutic usage in a variety of conditions, which suggests its clinical utility in the treatment of moderate chronic pain [62], inflammation [63], seasonal affective disorder, and sleep disturbances [64]. Additionally, melatonin inhibits the physiological process of platelet aggregation as well as the release of ATP and serotonin [65], presenting a nocturnal variation in the sensitivity of human platelets to melatonin [66]. The maximum effect of melatonin on platelet activity precedes the peak of melatonin concentration [65]. Higher expressions of fibrinogen and alterations in platelet distribution have been reported in fibromyalgia [67].

### 2.3. Implication of Melatonin in Improving the Alteration of Circadian Rhythms in FMS

Melatonin is involved in circadian rhythm synchronization and, consequently, in the regulation of fatigue and sleep–awake rhythms [68]. It also increases endogenous pain inhibition mechanisms and mood. As a result, the physiological processes that are sustained by melatonin availability are critical to the clinical symptoms of FMS, including cognitive impairment, exhaustion, persistent pain, and sleep disruption. Low levels of serotonin and tryptophan, which are precursors of melatonin, have been linked to a variety of symptoms in FMS patients, leading to the conclusion that melatonin may play a role in the etiology of FMS [1,69].

Although several studies have analyzed the disease from different points of views [70,71,72,73,74], based on the circadian ability of melatonin and its implication in the sleep–wake cycle, Zannette et al. designed a double-blind randomized controlled trial to assess indolamine in FMS patients [26].

Polysomnography is the ‘gold standard’ for sleep assessment, but activimetry has been used for more than 25 years to assess sleep–wake behavior and collect data on movement [75,76,77,78]. The American Sleep Disorders Association established the use of activimetry, which has an accuracy of 86%, as a valid method of evaluating specific domains of sleep research and sleep medicine [79]. Therefore, many researchers have used and are using this method in the field of FMS [80,81,82].

Some studies have shown that sleep–wake disorders, including early awakening, insomnia, non-restorative sleep, and poor sleep quality, in a high percentage of FMS patients [83] could be caused by defects in the production of tryptophan and serotonin, precursors of melatonin [84], which may explain a lower synthesis of melatonin during the night. In a longitudinal placebo-controlled design [77], after evaluating the chronobiological parameters of sleep, the most effective dose of melatonin (in terms of the total number of enhanced parameters) was in the range of 6 to 15 mg for 10 days, as an improvement was obtained in six out of seven sleep parameters analyzed (assumed sleep, immobility, real sleep time, sleep efficiency, sleep latency, and total nocturnal activity) when a dose of 15 mg of melatonin per day was administered [77]. Nevertheless, two pilot studies have proposed that the intake of 3 mg melatonin per day for 30 days and 6 mg/day for 10 days improved sleep disturbances [85,86]. Consequently, the differences found among the studies may be associated with the clinical profiles of the patients and the differences in the experimental design or duration of treatment (30–60 days versus 10 days). Table 1 summarizes these studies.

The results obtained with activimetry have been confirmed by those found through the subjective analysis of sleep using the Pittsburgh sleep quality index (PSQI). This questionnaire is widely used to measure sleep quality and disorders [88,89,90]. The Spanish version of this questionnaire has been shown to be an effective instrument for measuring the subjective perception of sleep quality in Spanish patients with FMS [90]. Castaño et al. showed that the perception of patients in relation to their quality of sleep was positive after taking 6 mg of melatonin for 10 or 15 days [77,78], and they agreed with other studies [87]. As the melatonin dose was increased, better results were observed in terms of increased rest at night.

Importantly, the differences in the results obtained after evaluating sleep quality by activimetry and the PSQI could be due to the fact that melatonin serves as a chronobiotic chemical as well as being engaged in pain pathways. Melatonin levels in FMS patients have been found to be altered, with lower secretion during the dark hours and increased secretion during the day [91]. Changes in melatonin secretion alter the circadian rhythm and sleep architecture, which may exacerbate depressed symptoms, fatigue, and pain, while masking the ability to regulate them [92].

### 2.4. Effects of Melatonin on Pain and Sleep Quality in FMS

Melatonin was proposed as a potential treatment for FMS due to its analgesic, anxiolytic, and chronotropic characteristics [33,52,84,85]. Previously, melatonin has been shown to be effective in the treatment of sleep disturbances in major depressive disorder, when it is used in a slow-release fashion alongside standard antidepressant treatment with fluoxetine [93]. On the other hand, ramelteon, which has high selectivity for MT1/MT2 receptors and little affinity for other receptors, such as opioids, among others, has been useful for insomnia symptoms in generalized anxiety disorder [94]. Even in seasonal affective disorder (SAD), such as winter depression, which has a circadian component, melatonin administration has been recommended in the afternoon, alongside the bright morning light, not only for treating sleep alterations due to circadian rhythm disorders but also because of an antidepressant effect [95]. It seems reasonable to propose new studies to evaluate the therapeutic indication of melatonin in nonseasonal depression, as well as other sleep and psychiatric disorders, in which a circadian component that might be present can be analyzed [95] for the implementation of melatonin treatment as a coadjuvant to FMS. It is possible that melatonin treatment of FMS may improve symptoms by regulating circadian rhythm synchronization and directly influencing pain pathways and/or the amounts of signaling molecules that govern pain [24,49,77,78]. If melatonin can improve pain in FMS patients, it is likely to result in greater sleep quality for them; thus, more randomized controlled trials are necessary to clarify this aspect.

The numerical pain scale, which is commonly used in clinical practice, revealed that the intensity of pain perceived by patients decreased remarkably after the intake of melatonin in a range of doses, from 3 mg/day to 15 mg/day for 10 days, specifically obtaining a dose-dependent effect [78]. Thus, the decrease in pain could be related to an increase in circulating melatonin, indirectly measured as urinary 6-sulfatoxymelatonin (aMT6-s), the major urinary metabolite of melatonin that accurately reflects nocturnal plasma melatonin [96]. The analgesic effects of melatonin have been confirmed in FMS patients, but with longer treatment periods: 3 mg/day for 30 days [85]; 5 mg/day for 8 weeks [86]; and 10 mg/day for 6 weeks [26].

Increased nociception may be linked to altered melatonin production in FMS patients, which can manifest clinically as hyperalgesia and/or allodynia [60,84,91]. Combination therapy trials using melatonin and fluoxetine [87] or amitriptyline [26] have offered further evidence of the efficacy of melatonin in the treatment of FMS and bolstered the need for additional research into other concomitant drugs. However, the data for combination therapy are insufficient to determine the optimal alternatives, and more research is needed to determine this possibility in FMS [12].

### 2.5. Relationship between Melatonin and Cortisol in Fibromyalgia Symptoms

The perception of improved mood and physical condition in patients with FMS, as determined by the score obtained on the visual analogue scale (VAS), was increased after melatonin (15 mg) compared to basal conditions [78]. Melatonin and cortisol are involved in the regulation of mood, the modulation of pain, and in anxiety [33,91,97]. The effects on mood could be attributed, at least in part, to the intake of melatonin. In general, patients with FMS present with high levels of anxiety and depressive symptoms that contribute to increasing the perception of pain and the somatization of symptoms, greatly impacting their quality of life [98]. After melatonin treatment, from 3 mg/day to 15 mg/day, ‘State-Anxiety’, that is, the anxiety that a subject temporarily has, was reduced compared to the anxiety that the subject presented at baseline [77]. However, ‘Trait-Anxiety’, i.e., the anxiety that is considered a latent trait of the subject, was unmodified. The ingestion of melatonin in these patients contributed to compensating for imbalances in the hypothalamic–pituitary–adrenal (HPA) axis [99], which also influences stress and anxiety levels [100,101].

In patients with FMS, alterations have been found not only in the pattern of melatonin secretion, but also in the serotonin and cortisol patterns [28]. At the physiological level, in healthy people, daytime melatonin concentrations are low and cortisol levels are high, and vice versa at night. In FMS patients, this relationship is reversed, so they generally have abnormally low cortisol levels during the day. Thus, melatonin treatment in these patients could have contributed to compensating for imbalances in the HPA axis. Recently, melatonin (9–15 mg for ten days) reduced, in a dose-dependent manner, urinary cortisol concentrations [78]. Although there is still disagreement regarding the cortisol levels in patients with FMS due to high variability in cortisol sample collection methods and different collection times, there is clearly dysregulation in the HPA axis in patients with FMS [102,103,104,105,106]. When cortisol levels are measured in first-void morning urine, the levels that the person had during the previous night are determined. After evaluating urinary and serum cortisol levels, lower values in the morning have been obtained in patients with FMS compared to the control [107,108]. On the other hand, Mahdi et al. (2011) found elevated serum cortisol concentrations at night [28]. Castaño et al. observed improvements in anxiety levels, mood, pain, and quality of life after the administration of melatonin [78]. Since melatonin exerts an inhibitory effect on the secretion of cortisol [109], it seems reasonable that restoring both altered levels may improve chronodisruption. With the administration of melatonin at night, the data indicate that cortisol levels are low in the morning, which may be considered a promising result, because it means that the administration of exogenous melatonin contributed to decreasing the abnormally high levels of cortisol at night in these patients.

Alternatively, to determine the effectiveness of an adequate treatment in this disease, not only must the benefits obtained in terms of anxiety, emotional state, pain, and/or sleep be evaluated, but also the effects it causes on the quality of life of a patient. In this sense, the fibromyalgia impact questionnaire (FIQ) is the most widely used. Castaño et al. revealed that the administration of 9, 12, and 15 mg of melatonin improved FIQ scores [78]. Notably, a short period of time (only 10 days for each dose) was required to observe the beneficial effects of melatonin. De Zanette et al. [26] demonstrated that 10 mg of melatonin per day for 6 weeks also improved FIQ scores. Other authors found significant decreases in total FIQ scores with lower doses of melatonin, e.g., 5 mg/day, but longer periods of administration, 8 weeks, were required [86].

The SF-36 questionnaire (SF-36) provides information on health-related quality of life and classifies it in different dimensions (physical function, social function, role—physical, role—emotional, vitality, body pain, mental health, and general health). Melatonin administration at 9 mg/day for 10 days has been shown to improve four of the eight dimensions evaluated by the SF-36. After 15 mg/day for 10 days, all of the evaluated dimensions improved remarkably. In other words, all of the patients declared that their emotional and health statuses, their social relationships, and their vitality improved [78]. The results of several questionnaires are summarized in Table 2.

As potent inhibitors of inflammation, glucocorticoids might be considered to have therapeutic effects in fibromyalgia [110]. The main clinical features (fatigue and pain) are associated with disturbed glucocorticoid receptor signaling pathways, rather than a decrease in glucocorticoid concentrations, since the reduction in glucocorticoid sensitivity is accompanied by increased fatigue frequency [110]. Fatigue, pain, headache, brain fog, mood and sleep disorders, among others, are symptoms commonly experienced in post-COVID-19 patients. These symptoms could be considered as manifestations of central sensitization [111]. In fact, fatigue is one of the core symptoms in central-sensitization-associated disorders [112], leading to the hypothesis that central sensitization might be an underlying common etiology in chronic pain patients and in patients with post-COVID-19 condition [113]. A common pathogenic mechanism is suggested because of the similarities among encephalomyelitis/chronic fatigue syndrome, FMS, and post-COVID syndrome [114].

In fibromyalgia, the balance between pro- and anti-inflammatory cytokines is suggested to be disrupted in favor of pro-inflammatory cytokines. Cytokines are released from both immune (monocytes, T cells, and macrophages) and non-immune cells (Schwann cells, fibroblasts, microglia, and astrocytes) [115,116]. Several gene variants are supposed to be associated with cytokine release and the inflammatory state in FMS [117,118,119].

### 2.6. Melatonin and Its Significant Antioxidant Role in FMS

A reduction in oxidative capacity has been proposed to be involved in the pathophysiology of FMS [120,121]. The oxidant/antioxidant balance may be important in this illness. The total urinary antioxidant capacity, determined in the serum of FMS patients, increased with melatonin doses of 9, 12, and 15 mg/day for 10 days, with a dose-dependent effect [77]. Taking into consideration the well known antioxidant properties of melatonin, it is possible that these effects may be attributed to elevated circulating melatonin concentrations in these patients. The largest antioxidant effects coincided with the largest increase in aMT6-s levels [77].

Furthermore, the identification of the melatonin binding site MT3 as the quinone reductase QR2 (NQO2) [122], a putative “receptor” present in mammalian brains [123], has seen increasing interest. In this way, this finding from the observation of a binding site [122] and via isothermal titration calorimetry [123] in both central and peripheral hamster tissues, as well as through the co-crystallization of the human enzyme NQO2 with melatonin [124,125], constitutes an original binding profile and very rapid kinetics of ligand exchange compared with the two melatonin G-coupled receptors, MT1 and MT2 [122]. The oxidoreductive properties of QR2 highlights the method for an incipient enzymatic investigation of the antioxidant function of melatonin, this enzyme being the fourth molecular target to explore the multiple facets of melatonin action, after MT1, MT2, and the transferase arylalkylamine N-acetyltransferase, which controls melatonin biosynthesis [122]. The possible capacity of MT3 binders to inhibit QR2 might be involved in several major pathological conditions, particularly the QR2-dependent production of radical oxygen species, which requires further exploration. It has been suggested that pharmacological melatonin concentrations, such as 100 µM and beyond, may inhibit QR2 activity, which is considered the previously named mitochondrial complex I, also referred to as NADH: ubiquinone oxidorreductase [126]. Therefore, the potential role of MT3/QR2 in mitochondrial dysfunction due to oxidative stress could be involved in the etiology of FMS.

### 2.7. Oxidative Stress and Neuroinflammation in FMS

Fibromyalgia can be considered as a central sensitization syndrome since its predominant pathogenic mechanism is the alteration of pain regulation at the brain level [127]. Neural cells are vulnerable to reactive oxygen species attacks and lipid peroxidation. Several studies have shown increased levels of oxidative stress markers in FMS [128,129]. Mitochondrial dysfunction, impaired bioenergetics, and reduced antioxidant enzyme levels are considered underlying factors of oxidative stress and inflammation in FMS. Central neuroinflammation, which is triggered by increased levels of cytokines and neurotrophic factors in cerebrospinal fluid, and central sensitization are closely connected in FMS [129]. Substance P, brain-derived neurotrophic factor, glutamate, nerve growth factor, and several inflammatory mediators activate glial cells, which produce pro-inflammatory cytokines which leads to neuroinflammation. This phenomenon increases the central processing of nociceptive input and contributes to chronic pain, allodynia, and hyperalgesia in FMS. As evidence of central neuroinflammation, the intrathecal concentration of IL-8 is elevated in fibromyalgia patients when compared to healthy controls [130], and it might be related to glial cell activation. The elevated IL-8 concentration with a lack of IL-1β increase indicates that symptoms in FMS are mediated by sympathetic nervous system, instead of the prostaglandin-related pathways [130,131]. Exogenous melatonin reduced levels of inflammatory markers, IL-1, 6, and 8, and TNF, and it has been proposed as a useful tool for the prevention and adjuvant treatment of inflammatory disorders [132].

### 2.8. Novel Findings in Therapies According to Animal Models of FMS

To estimate whether alternative and/or complementary medical treatments may improve the results of this disorder, recent studies have considered that pharmacological interventions provide variable benefits and common side effects [133].

Melatonin is able to maintain mitochondrial homeostasis and boost skeletal muscle resilience to damage by mending physiological levels of CoQ10 and other proteins (Figure 1). In addition, Suofu et al. [134] established that melatonin is produced in the mitochondria, the major site of free radical generation. This is extremely important in protecting these organelles and cells from damage due to the high capacity of indolamine. Moreover, melatonin has powerful neuroprotective qualities, including the ability to prevent mitochondrial cytochrome c release and subsequent caspase activation [135,136,137].

The potential positive effects of melatonin were stressed in reserpine-induced myalgic (RIM) rats whilst studying the processes related to the action of the indolamine. RIM rats exhibit FMS-like chronic pain symptoms and are excellent models for determining the etiogenesis of FMS and showing that mitochondrial dysfunction and oxidative stress, mediated by PGC-1, a main factor controlling mitochondrial biogenesis and shape; Mfn2, an outer mitochondrial membrane GTPase; and CoQ10, are implicated in FMS [138]. Several studies have found that RIM rats have decreased locomotor activity and body weight, as well as a considerable aversion to eating [139,140]. These findings are in line with the symptoms of FMS, including those in the revised fibromyalgia impact questionnaire (FIQR), as physical fitness has been observed to be associated with these symptoms [141,142]. Treatment with melatonin reduced FMS symptoms in RIM rats by supporting antioxidant responses in skeletal muscle and blood serum. Treatment of RIM rats with melatonin significantly improved their voluntary motor activity, increased both distance travelled and the rate of motor activity, and obtained values comparable to those in control rats. Long durations of inactivity, as in both RIM rats and FMS patients, cause changes in skeletal muscle, including increased production of ROS. This shows that oxidative stress may be a crucial factor in the development of muscle illness [138]. Many signaling pathways affecting muscle mass are regulated by mitochondria, and an imbalance in mitochondrial dynamics causes the formation of ROS and several other oxidative-associated factors, such as Mfn2 and PGC-1α [143,144]. Favero et al. found significant and moderate/strong expression of Mfn2 and PGC-1α, respectively, in control rats, despite their expression being drastically reduced (weak/very weak) in RIM animals [138]. After spontaneous exercise carried out daily, control rats exhibited a moderate/strong expression of myogenin, a transcriptional activator essential for the development of functional skeletal muscle in mice. Myogenin is, among other myogenic factors, the key player in the mechanisms of prenatal and postnatal myogenesis [145]. Consequently, the above-mentioned factors are broadly recognized for their contributions to maintaining muscle mass in addition to guaranteeing muscle regeneration and hypertrophy during the rodent life span [146].

In light of the information presented above, mitochondria are dynamic organelles that are critical for maintaining protein homeostasis in a variety of tissues, including skeletal muscle, in both health and sickness [147]. To verify these results, Favero et al. studied the expression of another marker of mitochondrial function, CoQ10 [138], which is not expressed in FMS patients [148,149,150]. The data obtained confirmed reduced CoQ10 expression in the skeletal muscle of RIM rats compared to control animals, implying that supplementation could alleviate the clinical symptoms associated with this illness [151]. Even CoQ10 supplementation provides a benefit in terms of relieving the sensation of pain in pregabalin-treated FM patients, possibly due to reduced mitochondrial oxidative stress and inflammation [152]. Figure 1 illustrates the hypothesized crucial function of mitochondria in FMS and the biosynthetic pathway mediated by PGC-1α.

The importance of melatonin in mitochondrial homeostasis is based on the mitochondria generating huge amounts of ROS in eukaryotic cells [153,154,155], and because of the role played by melatonin in the regulation of glutathione disulfide (GSSG)/glutathione (GSH) equilibrium. The antioxidant effect of melatonin and its ability to increase GSH levels may be of great importance for mitochondrial physiology by reducing the mitochondrial damage caused by free radicals and decreasing the loss of electrons in the inner mitochondrial membrane, where the electron transport chain, an oxide-reducing protein system formed by complexes I, II, III, and IV, resides [154,155]. In addition, melatonin has been shown to increase the number of mitochondria in cells when given long-term [156]. Experiments with radioactive melatonin reveal that this indolamine has binding sites in mitochondria [157]. Similarly, melatonin protects the brains of fetal rats against oxidant-mediated mitochondrial damage [158] and stimulates mitochondrial respiration in the livers of mice with accelerated senescence [159]. Alternatively, melatonin also exerts its protective action based on its ability to position itself between the polar heads of polyunsaturated fatty acids within cell membranes, consequently reducing lipid peroxidation and preserving optimal fluidity in the membranes [160,161,162,163]. A combined treatment of melatonin and folic acid, in a rat model of reserpine-induced fibromyalgia, may be useful in the treatment of FMS, thanks to its ability to target all mediators that contribute to the perpetuation of pain, from mastocytosis and related pro-inflammatory, vasoactive and neuro-sensitizing mediators to oxidative stress processes [164].

Therefore, the strong point of melatonin is caused by its higher efficiency in mitochondria compared to several kinds of antioxidants that have limited access to the same organelle. Ramis et al. [165] used a similar strategy, claiming that mitochondria-targeted antioxidants aggregate within the mitochondria at hundreds of times higher quantities and protect these vital organelles from oxidative damage.

## 3. Conclusions

Fibromyalgia is a chronic disease that leads to bouts of pain, which can be triggered by overexertion, mood disorders, such as states of anxiety or depression, and sleep disturbances. Despite having a benign character, because it does not produce physical sequelae, nor does it influence the patient’s survival, the impact it causes on the quality of life can be limiting. It is very important to establish a firm diagnosis because it saves a pilgrimage in search of diagnoses or treatments and allows setting realistic goals. FMS has no cure, so the goal of treatment is to reduce pain and treat the accompanying symptoms, to improve the quality of life of these patients. In this way, pain relievers, muscle relaxants and antidepressants drugs which increase serotonin levels, can improve FMS symptoms.

Antioxidants that target the mitochondria, such as melatonin, have scientific value and should be considered for enhancing mitochondrial health and/or disorders associated with the mitochondria. However, the main mechanism by which melatonin exerts analgesic effects is still unclear. Before evaluating the clinical applications of melatonin in the prevention and/or treatment of FMS in humans, a thorough understanding of the underlying mechanisms of its observed effects in nociception is required. FMS continues to have an unknown etiology, and this field of study progresses slowly. However, future studies, such as those with the mitochondria, and specifically those that focus on the mechanisms of neuroinflammation and central sensitization, could answer many questions and continue to support the potential of melatonin as an adjuvant molecule in fibromyalgia, taking into account the close relationship between melatonin, mitochondrial oxidative stress balance, and the proper integrated functioning of the nervous system.

## Figures and Tables

**Figure 1 biomedicines-11-01964-f001:**
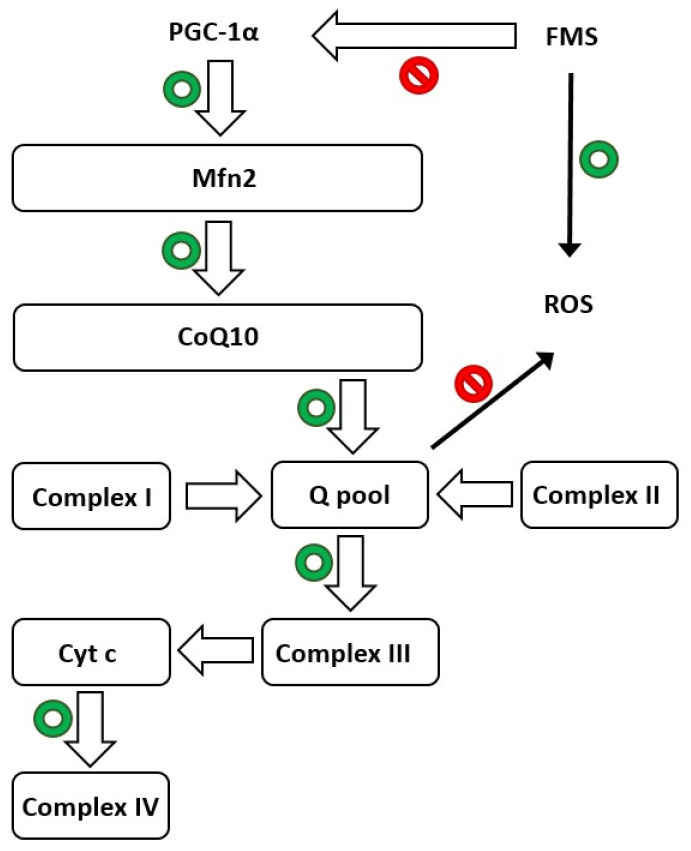
Schematic depicting mitochondria’s crucial function in normal life (green circles) and fibromyalgia syndrome (FMS), highlighting the biosynthesis route mediated by PGC-1α. Mfn2: mitofusin2; PGC-1α: peroxisome proliferator-activated receptor gamma coactivator 1-alpha; cyt c: cytochrome c; and CoQ10: coenzyme Q10. Red circles indicate inhibition of the pathway.

**Table 1 biomedicines-11-01964-t001:** Clinical trials of melatonin as therapy for fibromyalgia.

Study	*n*	Dose	Study Design	Results
Citera et al., 2000 [85]	19	3 mg	Pilot	Tender points, pain intensity, and sleep quality improved after 4 weeks
Acuña-Castroviejo et al., 2006 [86]	4	6 mg daily	Pilot	Pain and fatigue improved. Sleep–wake cycle normalized after 15 days
Hussain et al., 2011 [87]	101	5 mg daily/ 3 mg + 20 mg fluoxetine daily/5 mg + 20 mg fluoxetine daily	Double-blind RCT ^1^	In every case, FIQ ^2^ score improved. After 8 weeks, combination with fluoxetine was more effective
De Zanette et al., 2014 [26]	63	10 mg daily/10 mg + 25 mg amitriptyline daily	Double-blind RCT ^1^	On pain and threshold, melatonin was more helpful than amitriptyline alone. Tender points and sleep quality were improved after 6 weeks
Castaño et al., 2018 [77]	33	3, 6, 9, 12, and 15 mg daily	Longitudinal placebo-controlled design	Melatonin doses of 12–15 mg and 6–15 mg enhanced sleep quality, respectively. Melatonin (9–15 mg) for 10 days improved total antioxidant capacity
Castaño et al., 2019 [78]	36	3, 6, 9, 12, and 15 mg daily	Longitudinal placebo-controlled design	After 10 days, pain, happiness, quality of life, anxiety, and FIQ ^2^ improved

^1^ RCT: randomized controlled trial; ^2^ FIQ: fibromyalgia impact questionnaire.

**Table 2 biomedicines-11-01964-t002:** Effect of the administration of different doses of melatonin for 10 days on various indices.

Questionnaire	Melatonin (mg)				
	3	6	9	12	15
PSQI ^1^	n.s.	+	+	++	+
NPS ^2^	n.s.	n.s.	+	+	+
FIQ ^3^	n.s.	n.s.	+	+	+
SF-36 ^4^	n.s.	n.s.	+	+	+

^1^ PSQI: Pittsburgh sleep quality index; ^2^ NPS: numerical pain scale; ^3^ FIQ: fibromyalgia impact questionnaire total score; and ^4^ SF-36: short form-36 health survey subscales of general health. + indicates a significant improvement compared to the basal value, ++ indicates a significant improvement compared to the basal value as well as compared to 3 mg treatment, and n.s. indicates no significance.

## Data Availability

Not applicable.

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
