# Peer review of "Melatonin as a Coadjuvant in the Treatment of Patients with Fibromyalgia"

_biomedicines, 2023, doi:10.3390/biomedicines11071964_

Round 1

Reviewer 1 Report

Treating patients with fibromyalgia is not a simple task for a medical practitioner. On the one hand, treatment regimens are defined, but in some patients it is necessary to modify pharmacotherapy.
Interesting work, the authors conducted a thorough review of the literature.
I have a few comments and suggestions aimed at increasing the value of the work for the readers.
- The mechanisms of action are discussed very sparingly in the introduction. Please add a chapter on receptors, activated extra-receptor event systems. It would also be a good idea to compare different activation schemes and discuss interactions with melatonin receptor pathways.
- Please pay attention to platelet activity, what is the effect of melatonin especially as patients can be treated with SSRIs and similar drugs.
- please discuss the potential interactions of melatonin with groups of drugs used in patients with fibromyalgia.
- I don't like the conclusion. A suggestion appears, followed by "however" a moment later. Please remember that the papers will also be read by clinicians and here it is necessary to mark (perhaps in a small subsection) suggestions on when and how to start treatment. Who can you expect the best results from? I don't want to suggest that you give only an optimistic view, but please, in conclusion, discuss your own summary of use. It would be good to place before the conclusion chapter a chapter with a discussion on whom it should not be used and a general critical view resulting from previous research.

Author Response

Response to Reviewer 1 Comments

Comments and Suggestions for Authors

Treating patients with fibromyalgia is not a simple task for a medical practitioner. On the one hand, treatment regimens are defined, but in some patients it is necessary to modify pharmacotherapy.
Interesting work, the authors conducted a thorough review of the literature.
I have a few comments and suggestions aimed at increasing the value of the work for the readers.

Dear Reviewer 1, thank you very much for all your indications. We show you all the changes made in this document:

Point 1: The mechanisms of action are discussed very sparingly in the introduction. Please add a chapter on receptors, activated extra-receptor event systems. It would also be a good idea to compare different activation schemes and discuss interactions with melatonin receptor pathways.

Response 1: Thank you for this appreciation. We have considered this and have added a paragraph since line 122-47, incorporating your recommendation, mentioning the MT1 and MT2 agonist receptors, and MT3 as the quinone reductase; the nuclear receptors in the first part of the paragraph. In the second part, we have focused on the interactions with other receptors, most of all in relation to fibromyalgia symptoms.

Point 2: Please pay attention to platelet activity, what is the effect of melatonin especially as patients can be treated with SSRIs and similar drugs.

Response 2: Thank you for warning us about this important fact. We have added a sentence indicating this physiological effect in platelet aggregation in line 183-9.

Point 3: please discuss the potential interactions of melatonin with groups of drugs used in patients with fibromyalgia.

Response 3: In response to your request, we have remarked the treatment with fluoxetine or amitriptyline in line 290-3, and considered the treatment of the sleep disturbances in depressive, anxiety and seasonal affective disorders, in which melatonin has shown to be effective in line 261-5. Also we have added two more treatment for FMS: one is suplementation with Coenzima Q in pregabalin treated FM patients (line 517-9), and another is a combined treatment of melatonin and folic acid (line 542-7).

Point 4: I don't like the conclusion. A suggestion appears, followed by "however" a moment later. Please remember that the papers will also be read by clinicians and here it is necessary to mark (perhaps in a small subsection) suggestions on when and how to start treatment. Who can you expect the best results from? I don't want to suggest that you give only an optimistic view, but please, in conclusion, discuss your own summary of use. It would be good to place before the conclusion chapter a chapter with a discussion on whom it should not be used and a general critical view resulting from previous research.

Response 4: Thank you for this appreciation-we have modified the conclusions, adding more information of the most important aspects of FMS, focusing on the main points to take into account, such as the clinical, diagnosis (more expanded in line 52-9) and treatment of the symptoms, in order to make a summary of use, and to highlight future research perspectives.

We sincerely thank you all the corrections you have done in this manuscript.

Reviewer 2 Report

Melatonin and fibromyalgia

This extensive review addresses melatonin and potential for effect in fibromyalgia (FM) which could be of interest to both clinicians as well as researchers in the area of inflammation.

The manuscript is lengthy and difficult to follow and in the end the reader does not emerge with a clear understanding of the state of the art of melatonin and FM. Unfortunately I also do not see this review as a balanced unbiased evaluation of the literature; many references quoted and statements represent “cherry picking”. Many statements are presented as fact, whereas the evidence is currently more tenuous in many areas. Rather than tediously nitpick individual statements, paragraphs, references, I suggest a major overhaul of the paper with the following points in mind:

1.       The narrative sections are long, should be substantially abbreviated,  and could be improved by dividing into subsections. Each section should end with a summary statement of a few sentences.

2.       In reading this paper there is not a clear distinction between preclinical and clinical study….with the reader being led to believe that many statements are substantiated facts in the clinical setting. Again, the sections could be clearly delineated into “preclinical study” , “clinical study”, summary section.

3.       The authors have taken considerable liberties with many statements that cannot be substantiated by the current evidence. A few examples are given as illustration:

a.       Line 21: melatonin reduction of pain , improved mood and sleep in FM….the studies in FM are heterogenous, variable dosing, different outcomes, variable study duration and 2/6 are pilot studies. This statement is therefore not justified as it now stands and throughout the paper should be edited to say that preliminary clinical study SUGGESTS that melatonin may have a role in the management of FM.

b.       Line 25-26, 41: mitochondrial dysfunction, etc and FM….again statement is too strong…some studies have suggested a link between FM and mitochondrial dysfunction.

c.       Line 41: Ref 4 does not support evidence of oxidative stress in FM..” Most biomarkers (IL-6, IL-8, TNF-α, TBARS and protein carbonyl) and BDNF did not differ significantly between patients and controls, but the IL-10 levels were higher in FM patients (adjusted p=0.041).

d.       line 50, “confirm the pathophysiological connection between FM and oxidative stress”. The references quoted DO NOT CONFRIM the statement quoted….rather are suggestive…also Ref 8 and 10 are reviews and not studies.

e.       Section 2. Beginning line 91: Role of melatonin in FM. The evidence presented is almost all preclinical, which must be clearly stated. Furthermore statements of “clinical utility”, line 124 are derived from reviews 44 and 46, intestinal explants 45, and report of sleep quality in 30 participants in a RCT of Jerte valley cherry products (that contain melatonin, but without any accurate measurement of melatonin content) 47. These references do not support the statement of clinical utility.

f.        Line 134: Ref 49 is criteria and is not a reference to support the statement “melatonin plays a role in the etiology of FM”

g.       Etc

4.       The manuscript ends abruptly, without any clear synthesis. Suggest that there should be a section added before conclusion summarizing the major findings and pointers to areas of uncertainty that require further study.

5.       The authors have focused entirely on the potential effects of oxidative stress as a pathophysiological explanation for FM and have failed to reference the emerging evidence for pain sensitization, neuroinflammation and the interaction with oxidative stress. As it now reads, the novice reader could be led to believe that oxidative stress is the main mechanism…this is misleading and must be corrected throughout the manuscript. Some reference to the emerging literature on neurophysiological effects of melatonin in long-COVID would be helpful.

6.       Conclusions on the effects of melatonin in the clinical setting with reference to studies are far too optimistic as many studies are small, eg Ref 57,  and most studies in Table 1….these studies perhaps suggest effect, but more formal high quality studies are needed. Also authors often reference reviews rather than specific studies to support statements of effect.  This is a misrepresentation of the current evidence for effect.

Author Response

Response to Reviewer 2 Comments

Comments and Suggestions for Authors: Melatonin and fibromyalgia

This extensive review addresses melatonin and potential for effect in fibromyalgia (FM) which could be of interest to both clinicians as well as researchers in the area of inflammation.

The manuscript is lengthy and difficult to follow and in the end the reader does not emerge with a clear understanding of the state of the art of melatonin and FM. Unfortunately I also do not see this review as a balanced unbiased evaluation of the literature; many references quoted and statements represent “cherry picking”. Many statements are presented as fact, whereas the evidence is currently more tenuous in many areas. Rather than tediously nitpick individual statements, paragraphs, references, I suggest a major overhaul of the paper with the following points in mind:

Dear Reviewer 2, thank you very much for all your indications. We show you all the changes made in this document:

Point 1:

The narrative sections are long, should be substantially abbreviated, and could be improved by dividing into subsections. Each section should end with a summary statement of a few sentences.

Response 1: Thank you for this appreciation. We have abbreviated in some paragraphs, and added a subsection in line 122, and in line 435. We have also added a summary at the end of almost every section.

Point 2: In reading this paper there is not a clear distinction between preclinical and clinical study….with the reader being led to believe that many statements are substantiated facts in the clinical setting. Again, the sections could be clearly delineated into “preclinical study” , “clinical study”, summary section.

Response 2: We are agree with you. In fact, for this reason, we have considered to add more clinical studies, in addition to those already indicated in Table 2, for the treatment of sleep disturbances in mood disorders (line 256-65). Also in the end, we have added two more studies: one with CoQ10 supplementation in FM patients (line 518-20), and another with combined treatment of melatonin and folic acid in a rat model of reserpine-induced fibromyalgia (line 543-8).

Point 3: The authors have taken considerable liberties with many statements that cannot be substantiated by the current evidence. A few examples are given as illustration:

  1. Line 21: melatonin reduction of pain , improved mood and sleep in FM….the studies in FM are heterogenous, variable dosing, different outcomes, variable study duration and 2/6 are pilot studies. This statement is therefore not justified as it now stands and throughout the paper should be edited to say that preliminary clinical study SUGGESTS that melatonin may have a role in the management of FM.

Response 3a: We agree with you, and in line 21, we have written as you comment: “Preliminary clinical studies suggest that melatonin…”

  1. Line 25-26, 41: mitochondrial dysfunction, etc and FM….again statement is too strong…some studies have suggested a link between FM and mitochondrial dysfunction.

Response 3b: We have added a softer statement in line 25-26.

  1. Line 41: Ref 4 does not support evidence of oxidative stress in FM..”Most biomarkers (IL-6, IL-8, TNF-α, TBARS and protein carbonyl) and BDNF did not differ significantly between patients and controls, but the IL-10 levels were higher in FM patients (adjusted p=0.041).

Response 3c: Thank your for this observation. We have changed to another that better supports the affirmation made in line 41.

  1. line 50, “confirm the pathophysiological connection between FM and oxidative stress”. The references quoted DO NOT CONFRIM the statement quoted….rather are suggestive…also Ref 8 and 10 are reviews and not studies.

Response 3d: We thank you this comment, and have modified the references 8 and 10 for another more interesting due to analyze the imbalance between oxidants and antioxidants in FM patients.

  1. Section 2. Beginning line 91: Role of melatonin in FM. The evidence presented is almost all preclinical, which must be clearly stated. Furthermore statements of “clinical utility”, line 124 are derived from reviews 44 and 46, intestinal explants 45, and report of sleep quality in 30 participants in a RCT of Jerte valley cherry products (that contain melatonin, but without any accurate measurement of melatonin content) 47. These references do not support the statement of clinical utility.

Response 3e: In relation to your indication, we have added a new subsection in relation to melatonin receptors and its interactions with other receptors involvement in fibromyalgia symptoms, for presenting the information in a better way (line 124-onwards). The references 44-47 cited have changed for anothers to support the statement of clinical utility.

  1. Line 134: Ref 49 is criteria and is not a reference to support the statement “melatonin plays a role in the etiology of FM”

Response 3f: We apologize because this reference should actually be at the beginning, in line 52, corresponding to reference 1. Instead of this reference 49, we have added a new one.

Point 4: The manuscript ends abruptly, without any clear synthesis. Suggest that there should be a section added before conclusion summarizing the major findings and pointers to areas of uncertainty that require further study.

Response 4: To avoid this ending, we have rewritten the conclusions with a new paragraph at the beginning and with several changes at the end.

Point 5: The authors have focused entirely on the potential effects of oxidative stress as a pathophysiological explanation for FM and have failed to reference the emerging evidence for pain sensitization, neuroinflammation and the interaction with oxidative stress. As it now reads, the novice reader could be led to believe that oxidative stress is the main mechanism…this is misleading and must be corrected throughout the manuscript. Some reference to the emerging literature on neurophysiological effects of melatonin in long-COVID would be helpful.

Response 5: We agree with you, and have tried to apport more information to support the interaction of neuroinflammation and pain sensization with oxidative stress in several paragraphs (line 373-onwards; line 435-onwards). We thank you for this point of view in suggesting adding references on the effects of melatonin in prolonged COVID-19. We find it very interesting as well, and we have added this information in line 377-85.

Point 6: Conclusions on the effects of melatonin in the clinical setting with reference to studies are far too optimistic as many studies are small, eg Ref 57, and most studies in Table 1….these studies perhaps suggest effect, but more formal high quality studies are needed. Also authors often reference reviews rather than specific studies to support statements of effect. This is a misrepresentation of the current evidence for effect.

Response 6: We have apported more original studies intead of reference reviews. We also recognise that effectively more formal high quality studies are needed, most of all because there are a lot of unknowns regarding the mechanism by which these patients suffer from pain, in addition to sleep and mood disorders. We have tried to justify the potential role of melatonin as an adjunctive treatment in this disorder.

We sincerely thank you all the corrections you have done in this manuscript.

Reviewer 3 Report

This manuscript reviews current understanding and on-going studies about the pharmacological effect of melatonin as coadjuvant in the treatment of patients with fibromyalgia . Inclusion of wide range of research papers resulted in more than 124 citations, which is very helpful for the readers. The manuscript is concisely and well written. Authors also explore potential mechanism of melatonin in  mitochondria's function mitofusin2/CoQ-related in the pahtopsysiology of of fibromyalgia. Some minor comments were pointed as followings, in order to improve it:

57-58 and 71-77: the circadian hypothesis of anxio-depressive disorders should be here mentioned and this part this part about FMS/depression/MTL could be expanded here. Some non exaustive literature could be:

- https://doi.org/10.1016/j.bbr.2021.113724

- 10.1111/gbb.12369

- https://doi.org/10.1074/jbc.M114.559542

- https://doi.org/10.1038/nrd3140

- https://doi.org/10.1016/j.euroneuro.2005.09.002

106-113:  the interaction between the melatonergic and opioidergic systems has been investigated in 2 recent papers from Gobbi's group: MT2-KO mice have been found to have increaed endogenous opioid PENK expression in the RVM (10.1111/jpi.12671) and MT2 analagesic effects is mediated by MOR in the PAG-RVM descending pathway (10.1111/jpi.12825). Please add and discuss the melatonin/opioid interaction here. Ref 34 can be removed here.

187-195: Previous work showed that non selective MT1 and MT2 agonist melatonin and ramelteon per se has no antidepressant activity (Dolberg et al., 1998, https://doi.org/10.1176/ajp.155.8.1119 ; Gross et al., 2009, https://doi.org/10.5664/jcsm.27389 ), but it is effective in  patients suffering from seasonal affective disorder (Lewy et al., 2006, PNAS, 103 (2006), pp. 7414-7419). How this could be related to FMS?

- 301 "2.3. Melatonin and its significant antioxidant role in FMS". Interstingly, the MT3 receptor is a quinone redictase-2 (10.1074/jbc.M005141200 ; 10.1111/j.1600-079X.2007.00513.x ). Although there is limited literature about this melatonin recpetor subtype (see MT3: The Other Melatonin Binding Site, capt 5 in: New Developments in Melatonin Research, 2013 Nova Science Publishers, Inc, ISBN: 978-1-62618-843-3,), it would be interesting to mention and discuss this the potential role of the MT3/quinone redictase-2 as in mitochondrial dysfunction/oxidative stress in FBS.

Author Response

Response to Reviewer 3 Comments

Comments and Suggestions for Authors

This manuscript reviews current understanding and on-going studies about the pharmacological effect of melatonin as coadjuvant in the treatment of patients with fibromyalgia . Inclusion of wide range of research papers resulted in more than 124 citations, which is very helpful for the readers. The manuscript is concisely and well written. Authors also explore potential mechanism of melatonin in mitochondria's function mitofusin2/CoQ-related in the pahtopsysiology of of fibromyalgia. Some minor comments were pointed as followings, in order to improve it:

Dear Reviewer 3, thank you very much for all your indications. We show you all the changes made in this document:

Point 1: 57-58 and 71-77: the circadian hypothesis of anxio-depressive disorders should be here mentioned and this part this part about FMS/depression/MTL could be expanded here. Some non exaustive literature could be:
- https://doi.org/10.1016/j.bbr.2021.113724
- 10.1111/gbb.12369
- https://doi.org/10.1074/jbc.M114.559542
- https://doi.org/10.1038/nrd3140
- https://doi.org/10.1016/j.euroneuro.2005.09.002

Response 1: Thank you for this appreciation. We have considered this important aspect and have added the circadian hypothesis of anxio-depressive disorders in line 96-onwards, incorporating your recommendation about expanding FMS/depression/Melatonin in line 256-70.

Point 2: 106-113: the interaction between the melatonergic and opioidergic systems has been investigated in 2 recent papers from Gobbi's group: MT2-KO mice have been found to have increaed endogenous opioid PENK expression in the RVM (10.1111/jpi.12671) and MT2 analagesic effects is mediated by MOR in the PAG-RVM descending pathway (10.1111/jpi.12825). Please add and discuss the melatonin/opioid interaction here. Ref 34 can be removed here.

Response 2: Thank you for this interesting comment because there is no doubt that the melatonin/opioid interaction seems to be very important in the treatment of FM. We have added in line 150-3 the involvement of mu opiod receptor and MT2 receptors through modulation of descending antinociceptive pathways in the periaqueductal gray of the brainstem, and changed that reference for your reference suggested. Also, we have added the other reference recommended for discussing the melatonin/opioid interaction, in line 163-73.

Point 3: 187-195: Previous work showed that non selective MT1 and MT2 agonist melatonin and ramelteon per se has no antidepressant activity (Dolberg et al., 1998, https://doi.org/10.1176/ajp.155.8.1119 ; Gross et al., 2009, https://doi.org/10.5664/jcsm.27389 ), but it is effective in patients suffering from seasonal affective disorder (Lewy et al., 2006, PNAS, 103 (2006), pp. 7414-7419). How this could be related to FMS?

Response 3: In response to your request, we have reported the results obtained with melatonin in the improvement of sleep disturbance in the major depressive disorder, and in generalized anxiety disorder, and its efectiveness in seasonal affective disorder (SAD) as antidepressant effects. Because the SAD shares similar characteristics in common with FM, such as the perturbed circadian rhythms and sleep, as well as alterations in melatonin secretion, it may be suggested that this indication of melatonin as coadjuvant can be implented to FM, in order to improve the symptoms and sleep quality (line 108-11).

Point 4: 301 "2.3. Melatonin and its significant antioxidant role in FMS". Interstingly, the MT3 receptor is a quinone redictase-2 (10.1074/jbc.M005141200 ; 10.1111/j.1600-079X.2007.00513.x ). Although there is limited literature about this melatonin recpetor subtype (see MT3: The Other Melatonin Binding Site, capt 5 in: New Developments in Melatonin Research, 2013 Nova Science Publishers, Inc, ISBN: 978-1-62618-843-3,), it would be interesting to mention and discuss this the potential role of the MT3/quinone redictase-2 as in mitochondrial dysfunction/oxidative stress in FBS.

Response 4: Thank you for this great apportation. We have mentioned in line 413-33 this potential rol of MT3/quinone reductase-2 in FMS, due to mitochondrial dysfunction produced by oxidative stress, being considered the mitochondrial complex I.

We sincerely thank you all the corrections you have done in this manuscript, and for the high level of knowledge in the issue as well as in the way of transmitting it.

Round 2

Reviewer 1 Report

the manuscript has been significantly expanded, so I believe it may be considered for publication

Author Response

Dear Reviewer, thank you very much for your kind consideration.

Reviewer 2 Report

Thank you to the authors for response to my comments.

Once again, I note that the manuscript has not been sufficiently edited as I suggested. The narrative and paragraphs are long and should be improved by identifying subsections as well as reduction in length as there is considerable redundancy.

The manuscript should also be edited by a native English speaker to improve the language.

Once again, authors make statements about evidence from “studies”, and use narrative reviews  rather than  specific studies…this must be corrected throughout.

I have the following specific comments:

Line 59: reference should be 3

Line 107:  ref 24 addresses  a study of the endogenous effect of melatonin in FM and is not a reference for the various drug  treatments for FM…please supply an appropriate reference

Line 108: ref 25, 26 describe 2 individual drug studies…and do not reflect the statement “are not effective in fighting FMS”….rather reference some guidelines that are more generic.

Line 151: “allodynic effect”…this is incorrect and I believe should be analgesic effect

Line 159-161: this line is ambiguous and requires editing for clarification, as it stands it is not clear….? Should read “melatonin-induced anti-nociceptive effects.

Line 187: there is no evidence or data to support the statement that FM is a procoagulant condition..please remove this sentence.

Line  205: ref 33 and 49 are reviews and not “studies that highlight the benefits of treatment with melatonin in FM. Please edit this sentence to reflect the true evidence in FM, and remove the “referenced of reviews”.

Table 1: please make a statement in the discussion that of the 4 RCT’s with melatonin (Table has 6 but 2 are pilot studies), 2 by Castano are in fact the same study (97 patients screened with 36 and 37 recruited for each study) with report of sleep quality in one and QOL in the other. Furthermore, the other 2 studies in Table 1 report on a combination of melatonin with either amitriptyline or fluoxetine. Therefore these results are far from convincing and must be acknowledged.

Line 248-249: please reference the statement of “altered melatonin levels in FM”

Line 271-272: much too strongly positive…in view of the considerable biases of studies the studies have NOT CONFIRMED that melatonin is a positive therapy for FM…there is merely a suggestion that there may be impact on multiple symptoms, but with a need for well-designed RCT’s. Please edit appropriately.

Line 278: Reference 77 addresses only sleep quality, there is not a single measurement of pain as stated by the authors., perhaps they refer to Ref 78.

Line 464. Title 3. Novel finding in FMS therapies. The narrative does not follow on the title. This section is mostly addressing muscle changed in animal models of FM related to activity, also with much repetition regarding ROS and mitochondrial homeostasis. It is also not a stand alone section and should be identified as 2.6.

Author Response

Response to Reviewer 2 Comments

We thank the reviewer for the kindness in reviewing our manuscript, as well as  for the comments and suggestions for improvement, which have been incorporated.

Thank you to the authors for response to my comments.

Once again, I note that the manuscript has not been sufficiently edited as I suggested. The narrative and paragraphs are long and should be improved by identifying subsections as well as reduction in length as there is considerable redundancy.

To reduce the length and the redundancy of the manuscript, we have deleted several sentences through the text of the document, which leads us to shorten the length of the manuscript by almost a page.

The manuscript should also be edited by a native English speaker to improve the language.

We are very sorry if some of the expressions were wrong or difficult to follow. The manuscript was edited by a Company for general proofreading and editing (San Francisco Edit, invoice No: 220728) and numerous language deficiencies have been corrected.

Once again, authors make statements about evidence from “studies”, and use narrative reviews rather than specific studies…this must be corrected throughout.

We apologize if in several cases we have referred to reviews instead of specific studies. We have modified it to the best of our ability.

I have the following specific comments:

Dear Reviewer 2, thank you very much for all your indications. We show you the specific changes made in this document:

Point 1:

Line 59: reference should be 3

Response 1: Thank you for this indication. We have changed correctly the reference 3 in line 59.

Point 2: Line 107: ref 24 addresses a study of the endogenous effect of melatonin in FM and is not a reference for the various drug treatments for FM…please supply an appropriate reference

Response 2: The reference 24 has been removed of place, and we have better explained the propose of this work in line 106-7.

 Point 3: Line 108: ref 25, 26 describe 2 individual drug studies…and do not reflect the statement “are not effective in fighting FMS”….rather reference some guidelines that are more generic.

Response 3a: For better understanding, we have improved this sentence in line 107-9, and we have supressed the expression you mentioned above.

Point 4: Line 151: “allodynic effect”…this is incorrect and I believe should be analgesic effect.

Response 4: We apologize for this mistake. It should be written as “antiallodynic effect”. We have corrected it in line 152.

Point 5: Line 159-161: this line is ambiguous and requires editing for clarification, as it stands it is not clear….? Should read “melatonin-induced anti-nociceptive effects.

Response 5: We totally agree with you, and we have written in line 161, as you have indicated: “melatonin-induced anti-nociceptive effects”.

Point 6: Line 187: there is no evidence or data to support the statement that FM is a procoagulant condition..please remove this sentence.

Response 6: We appreciate your comment and have removed this sentence in line 187, and have added the expression “have been reported” after the sentence “higher expression of fibrinogen and alterations in platelet distribution in fibromyalgia”.

 Point 7: Line 205: ref 33 and 49 are reviews and not “studies that highlight the benefits of treatment with melatonin in FM. Please edit this sentence to reflect the true evidence in FM, and remove the “referenced of reviews”.

Response 7: We are totally agree with you on this point, because this sentence needs to be edited. Thus, we have rewritten this sentence in line 205-6, and also removed the reference to reviews, leaving the randomized controlled trial of Zannette et al.

 Point 8: Table 1: please make a statement in the discussion that of the 4 RCT’s with melatonin (Table has 6 but 2 are pilot studies), 2 by Castano are in fact the same study (97 patients screened with 36 and 37 recruited for each study) with report of sleep quality in one and QOL in the other. Furthermore, the other 2 studies in Table 1 report on a combination of melatonin with either amitriptyline or fluoxetine. Therefore these results are far from convincing and must be acknowledged.

Response 8: Thank you for your clarifications. We have focused on detailing the type of the study in each case (lines 205, 218, 225). On the other hand, the parameters of sleep studied by the longitudinal placebo-controlled design by Castaño et al, 2018 [77] are mentioned in lines 218-24, while the second study of Castaño et al 2019 [78] refers about pain levels (line 273-6 and 298-300), cortisol concentration decreased (line 323-5 and 334), FIQ (Fibromyalgia Impact Questionnaire) (line 344-6), quality of life (line 356-8).

 Point 9: Line 248-249: please reference the statement of “altered melatonin levels in FM”

Response 9: We apologize for it. We have included the reference in the text (line 249).

 Point 10: Line 271-272: much too strongly positive…in view of the considerable biases of studies the studies have NOT CONFIRMED that melatonin is a positive therapy for FM…there is merely a suggestion that there may be impact on multiple symptoms, but with a need for well-designed RCT’s. Please edit appropriately.

Response 10: We thank you this comment. In order to answer to your request, we have supressed the paragraph aboved indicated, and written in line 272 “more randomized controlled trials are necessary to clarify…”. Also we have added in line 267-8 “It is possible that melatonin treatment of FMS may improve symptoms” and “melatonin could improve pain in FMS patients” in line 270, for making just a suggestion.

 Point 11: Line 278: Reference 77 addresses only sleep quality, there is not a single measurement of pain as stated by the authors., perhaps they refer to Ref 78.

Response 11: We thank you this good observation. We have revised this reference and objetived that this change is necessary. It appears in line 276 as reference 78 instead 77.

 Point 12: Line 464. Title 3. Novel finding in FMS therapies. The narrative does not follow on the title. This section is mostly addressing muscle changed in animal models of FM related to activity, also with much repetition regarding ROS and mitochondrial homeostasis. It is also not a stand alone section and should be identified as 2.6.

Response 12: This is a good point of view, more adequate than we did before. So we have added this tittle: “2.6. Novel findings in therapies according to animal models of FMS”, changing the number 3 for 2.6.

 We sincerely thank you all the corrections you have done in this manuscript, and the way of transmitting these observations that allowed us to learn during the correction process.

Round 3

Reviewer 2 Report

Thank you for the revision. I am satisfied with the edits to the scientific information, but remain dissatisfied with the style and structure of this paper for the following reasons:

1. Once again this review is lengthy, with much redundancy and has only been shorted by 24 lines, barely 1/2 page and not as stated by authors by a whole page.

2. I previously suggested that additional subheadings would facilitate the reader..this has not been done

3. Although authors state that manuscript was proofread for English, unfortunately the grammar and syntax is poor.

4. There are still some errors in reference numbering: Line 107-108 should be Ref 25, 26; line 109 should be reference 24; line 206 should be ref 24

Author Response

Response to Reviewer 2 Comments

We thank the reviewer for the kindness in reviewing our manuscript, as well as for the comments and suggestions for improvement, which have been incorporated.

Thank you for the revision. I am satisfied with the edits to the scientific information, but remain dissatisfied with the style and structure of this paper for the following reasons:

  1. Once again this review is lengthy, with much redundancy and has only been shorted by 24 lines, barely 1/2 page and not as stated by authors by a whole page.

To reduce the length and the redundancy of the manuscript, we have deleted several sentences through the text.

  1. I previously suggested that additional subheadings would facilitate the reader..this has not been done

Previously we added two subheadings (2.1. Melatonin receptors and its interactions with other receptores involvement in fibromyalgia and 2.5. Oxidative stress and neuroinflammation in FMS). Now, according to your indication, we have added another two subheadings: 2.2. Interactions of melatonin on receptors implicated in the pain in FMS and 2.4. Effects of melatonin on pain and sleep quality in FMS.

  1. Although authors state that manuscript was proofread for English, unfortunately the grammar and syntax is poor.

We have corrected several gramatical and syntax errors throughout the text. We have highlighted them in the new attached version.

  1. There are still some errors in reference numbering: Line 107-108 should be Ref 25, 26; line 109 should be reference 24; line 206 should be ref 24.

We appologize for these errors. We have corrected them in reference numbering, so that it corresponds to the text.

We sincerely thank you all the corrections you have done in this manuscript, and the way of transmitting these observations that allowed us to learn during the correction process.
